# ADAPTIVE COMPRESSION OF THE LATENT SPACE IN VARIATIONAL AUTOENCODERS

## ABSTRACT

Variational Autoencoders (VAEs) are powerful generative models that have been widely used in various fields, including image and text generation. However, one of the known challenges in using VAEs is the model's sensitivity to its hyperparameters, such as the latent space size. This paper presents a simple extension of VAEs for automatically determining the optimal latent space size during the training process by gradually decreasing the latent size through neuron removal and observing the model performance. The proposed method is compared to traditional hyperparameter grid search and is shown to be significantly faster while still achieving the best optimal dimensionality on four image datasets. Furthermore, we show that the final performance of our method is comparable to training on the optimal latent size from scratch, and might thus serve as a convenient substitute.

## 1 INTRODUCTION

Variational Autoencoders (VAEs) (Kingma & Welling, 2014) are generative models that learn a probabilistic mapping between the data and the latent space, which allows for the generation of new, unseen data samples that are similar to the training data. Although VAEs have proven to be highly effective in this task, they are known for their sensitivity toward individual hyperparameters (Hu & Greene, 2018) and particularly the size (dimensionality) of the latent space. The selection of optimal latent dimensionality in VAEs is a crucial step in achieving the desired trade-off between reconstruction quality and clustering capability. Several works have studied and tried to mitigate this problem (Higgins et al., 2016), (Kim & Mnih, 2018) (Shao et al., 2020), however, in most current models, the optimal latent dimensionality is still chosen based on the specific task and data at hand.

Traditionally, determining the optimal latent space size has been done through a process of trial and error or by manually tuning the hyperparameters through grid search and observing the model's performance. However, this process can be time-consuming (as there are several hyperparameters to tune in parallel) and unreliable, making VAEs less practical for real-world applications (Way et al., 2020),(Ahmed & Longo, 2022).

In this paper, we propose ALD-VAE (Adaptive Latent Dimensionality) - a novel, automated approach for determining the optimal latent space size in VAEs. Our approach involves gradually decreasing the latent space size by removing neurons during the training process and observing the reconstruction loss, Fréchet Inception Distance (FID) and cluster quality of the latent samples using the Silhouette score.

We show that the ALD-VAE approximates the optimal number of dimensions and thus avoids the need for manual tuning, plus reaches the same accuracy as if it was trained on the optimal latent dimensionality from scratch. We demonstrate the effectiveness of our approach on four image datasets ranging from toy-level (SPRITES, MNIST) to real-world noisy data (EuroSAT). With our proposed method, VAEs can be made more practical for real-world applications and the need for manual tuning can be eliminated. The corresponding code and more detailed results can be found in the GitHub repository [1].

---

[1] GitHub URL (anonymized)

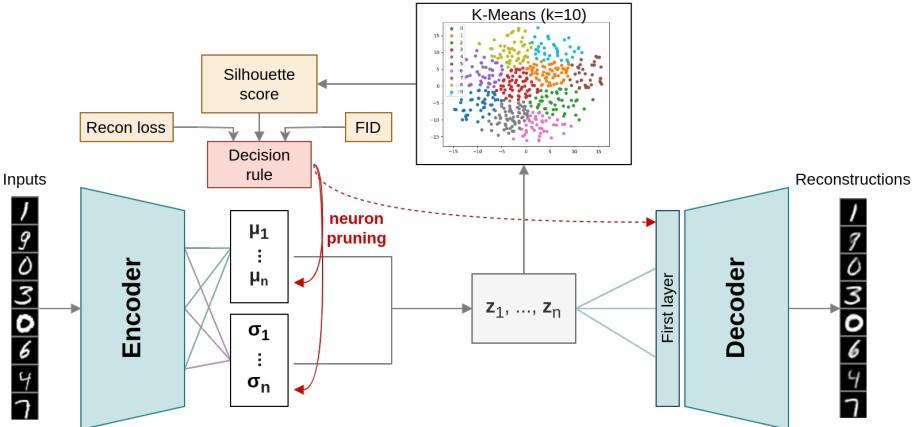

Figure 1: Overview of our training procedure with shrinking latent space size. After each epoch, we encode a batch of 500 random samples from the validation dataset and run K-means clustering. We then calculate the Silhouette score based on the detected clusters. The Decision rule observes the approximate slope of the Silhouette score, FID and the reconstruction loss over epochs. We prune 1-5 neurons after each 5 epochs from the encoder's $\mu$ and $\sigma$ layers and the decoder's first layer until the slopes meet our criteria (see Section 3.3.1), then the latent space size remains fixed.

## 2 RELATED WORK

Hyperparameter tuning in VAEs has become a common subject of research in recent years as it is usually a necessary step for obtaining optimal results on the chosen dataset (Locatello et al., 2019), but the tuning process can also influence the results of comparison between various VAE architectures (Hu & Greene, 2018).

Several studies provided a comparison of VAE performance under various latent space dimensionalities explored through grid search. For example, Ahmed & Longo (2022) examine the optimal latent space size for electroencephalographic (EEG) signals by comparing 25 different latent sizes, Way et al. (2020) compare different latent space dimensionalities for gene expression data. Considering the high computational costs of training multiple models with different hyperparameters for comparison, other works focused on hyperparameter tuning on the fly during a single training.

Mondal et al. (2020) propose a method to automatically estimate the optimal latent dimensionality in adversarial auto-encoders by introducing an additional trainable mask layer that learns to hide the least informative dimensions. However, this method requires adopting a separate optimizer for the mask layer and relies on a discriminator network which is not present in VAEs. A different approach is the two-stage VAE (Dai & Wipf, 2019b), where first a model with larger than optimal latent dimensionality is trained and then another VAE is finetuned during the second stage to learn the correct probability measure. Although close to our work, this approach does not focus on the clustering capability in VAEs.

In this paper, we focus on an adaptive decrease of the latent space size in VAEs during a single training which would be applicable to any VAE or its adaptation without requiring extra computational costs. We gradually find the optimal dimensionality based on the observation of the model's reconstructing and clustering capabilities in order to maximize the VAE's performance.

## 3 ADAPTIVELY COMPRESSING LATENT SPACE

In this section, we first formulate the problem of latent space compression and the task we are focusing on. Next, we describe the general VAE architecture. Finally, we define our proposed ALD-VAE training algorithm for automatic estimation of the optimal latent dimensionality.

## 3.1 PROBLEM FORMULATION

Let $\mathcal{D}$ be a dataset consisting of datapoints $\mathbf{x}$ lying in an $n$-dimensional manifold $\mathcal{X}$ embedded in $\mathbb{R}^d$. Variational autoencoders (Kingma & Welling, 2014) then establish a probabilistic relationship between $\mathcal{X}$ and a learned low-dimensional latent space $\mathcal{Z}$. It is our goal to find the optimal latent dimensionality $n_z$ within $\mathcal{Z}$ that matches the intrinsic dimensionality $n$ of the data. In real-world datasets, the intrinsic dimensionality $n$ has been shown to be much smaller than the dimensionality of $\mathbb{R}^d$, i.e. $n << d$ (Narayanan & Mitter, 2010), and using $n_z >> n$ might thus lead to redundant dimensions and noisy outputs. One of the metrics that can show the qualitative performance of the model is Fréchet Inception Distance (FID) (Heusel et al., 2017) and, as demonstrated by Dai & Wipf (2019a), too low as well as too high $n_z$ is non-optimal (see the typical u-shape of FID for different latent dimensionality e.g., in Fig.2).

One way to find the optimal $n_z$ would be thus by finding the minimum of FID. However, in many cases, it is also desirable to cluster the data in the latent space based on the mutual similarity between the data, e.g., for image datasets with multiple classes. Quality of the clusters in the latent space can be evaluated using Silhouette score $S = \frac{1}{x} * \sum_{i=1}^{x} s(i)$ calculated over $x$ samples encoded by the encoder network. For each sample $i$, $s_i$ can be computed as:

$$s_i = \frac{b_i - a_i}{max(b_i, a_i)} \tag{1}$$

where $b_i$ is the inter-cluster distance defined as the average distance to the closest cluster $C$ of datapoint $i$ (except for the one it is part of):

$$b_i = \min_{k \neq i} \frac{1}{|C_k|} \sum_{j \in C_k} d(i, j)$$

and $a_i$ is the intra-cluster distance defined as the average distance to all other points within the same cluster $C$:

$$a_i = \frac{1}{|C_i| - 1} \sum_{j \in C_i, i \neq j} d(i, j)$$

As shown in Fig. 2, the Silhouette score also changes with $n_z$, but can be optimal in different dimensionality than FID.

Our objective is thus to find such latent dimensionality $n_z$, that minimizes the FID score and reconstruction loss $\mathcal{L}_r$ (which is a part of the VAE objective loss function, see Sec. 3.2), while maximizing the Silhouette score $S$:

$$n_z = \arg\min_{n_z \in \mathbb{Z}} FID(n_z) \wedge \arg\min_{n_z \in \mathbb{Z}} \mathcal{L}_r(n_z) \wedge \arg\max_{n_z \in \mathbb{Z}} S(n_z), \tag{2}$$

## 3.2 VAE OBJECTIVE

As proposed by Kingma & Welling (2014), VAEs are trained end-to-end in an unsupervised manner: first, the encoder maps the input data into a distribution over the latent space, typically by using multiple layers of neural networks. Second, samples are drawn from the distribution and the decoder maps them back to the original data space. The objective of the VAE is to maximize the evidence lower bound (ELBO), which is equivalent to maximizing the likelihood of the data and regularizing the approximate posterior to be close to the prior. The ELBO loss term is denoted as follows:

$$\mathcal{L}_{ELBO} = \mathbb{E}_{q(z|x)} \log p(x|z) - KL(q(z|x)||p(z)) \tag{3}$$

where $\mathbb{E}_{q(z|x)} \log p(x|z)$ is the expected log-likelihood of the data under the approximated posterior distribution, $KL(q(z|x)||p(z))$ is the Kullback-Leibler divergence between the approximated posterior distribution and the prior distribution over the latent variables.

The size of the latent space is equal to the dimensionality of the multivariate distribution learned by the encoder. The parameters of the distribution (a vector of means, $\boldsymbol{\mu}$, and a vector of variances, $\boldsymbol{\sigma}$) are usually provided by the two fully connected output layers of the encoder network, each of them having the dimensionality $n_z$. Since the output size of the layer is conventionally decided at model initialization, the latent space remains fixed during training.

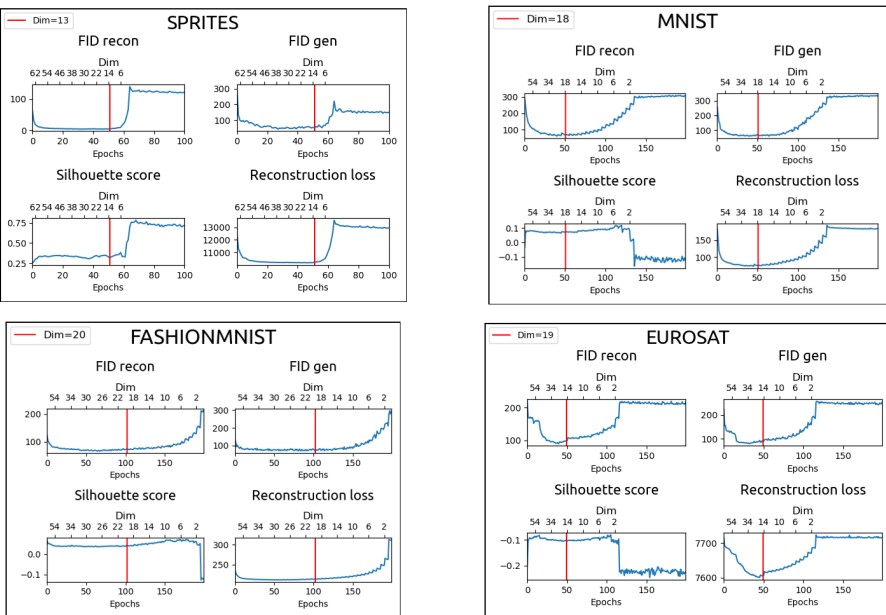

Figure 2: The values of the four observed metrics (FID for reconstructions, FID for generations, Silhouette score and Reconstruction loss) during training when the latent space is gradually reduced until Dim=2. The upper axis shows the dimensionality at the given epoch, the vertical red line shows when would our ALD algorithm stop the compression. We show the results for four datasets.

## 3.3 PROPOSED ALGORITHM

We strive to minimize the latent space dimensionality $n_z$ during the VAE training procedure to a value that is optimal in terms of the model's reconstruction accuracy and latent space regularization/clustering capability, i.e. the Silhouette score (see Eq. 2). In this section, we first describe the process of downsizing $n_z$ during training without the need of retraining from scratch, then we summarize the stopping mechanisms used to determine whether the adapted dimensionality is an optimal solution or not. For a high-level overview of the whole procedure, see Figure 1 or the pseudocode in Algorithm 1.

### 3.3.1 LATENT DIMENSIONALITY REDUCTION

As mentioned in Section 3.2, the latent space size $n_z$ is given by the output feature size of the two fully-connected layers $\boldsymbol{\mu}$ (learning the means of the multivariate distribution) and $\boldsymbol{\sigma}$ (learning the variances) which process the encoder's output.

The latent dimensionality reduction works as follows (see also Algorithm 1):

1. Start by choosing a random, high-enough initial dimensionality $n_z$.
2. Train the VAE model for $p$ epochs.
3. Check the stopping criterion (see Eq. 4). If stopping criterion is not met, proceed to 4., otherwise fix the current $n_z$ and continue training.
4. Remove $n$ neurons from each of the two last encoder layers of the VAE model and proceed to 2.

In each reduction process (step 4), we replace the last two fully-connected layers of the trained VAE encoder with new ones, which have the original output feature dimension reduced by $n$. We then copy the learned weights and biases from the old layer into the newly initialized layer and omit the $n$ values which are now redundant. We choose the weights and biases randomly, although a possible additional metric might be used for selection such as the dimension-specific KL divergence. As the latent vectors $z$ have a reduced dimensionality after downsizing the encoder layers, we perform the same operation on the first decoder network layer.

We perform the latent space compression reactively during the training. In the beginning, we remove $n = 5$ neurons (corresponds to the parameter *latent_decrease* in Alg. 1) after each $p = 5$ epochs (corresponds to the parameter patience $p$ in Alg. 1), until the slope of the Silhouette score over the last 10 epochs becomes positive (see Eq. 5 for the slope calculation). Afterwards, we slow down the pruning by reducing $n$ to 1 and prune only 1 neuron after each $p = 5$ epochs, so that we can slowly approach the optimal value. This is done iteratively until the stopping mechanism decides that the current dimensionality is optimal and $n_z$ remains fixed. Please note that although we choose the initial $n$ and $p$ empirically, we also evaluated the algorithm with different initial values and the final optimal dimensionality $n_z$ was similar, showing the robustness of the compression mechanism.

### 3.3.2 STOPPING MECHANISM

As demonstrated in Fig. 2, the Silhouette score, as well as FID and the reconstruction loss react to the latent space size throughout the training. We seek such latent space dimensionality $n_z$, under which we get the maximal Silhouette score ($S$) and minimal reconstruction loss ($-\mathbb{E}_{q(z|x)} \log p(x|z)$) as well as FID ($FID$). Since the Silhouette score has an optimum in different $n_z$ than reconstruction loss and FID, an optimal balance between them has to be found.

---

**Algorithm 1** Adaptively compressing latent space

---

1: **Input:** data $x_{train}$, $x_{val}$, number of data classes $k\_classes$
2: **Parameters:** init latent dims $n_z$, $latent\_decrease$, patience $p$, floating window $w$
3: s_scores, recon_losses, fid_g, fid_r ← empty list
4: $compressing \leftarrow 1$
5: **for** $epoch$ in $num\_epochs$ **do**
6:     train_loop($x_{train}$)
7:     val_loop($x_{val}$)
8:     $labels$ = KMeans.fit($z\_samples$, k=$k\_classes$)
9:     $s\_score$ = silhouette_score($z\_samples$, labels=$labels$)
10:     $recon\_loss$ = calculate_nll(model, data=$x_{val}$)
11:     $FID\_g$, $FID\_r$ = calculate_fid(model, data=$x_{val}$)
12:     append $s\_score$ to s_scores
13:     append $recon\_loss$ to recon_losses
14:     append $FID\_g$ to fid_g, $FID\_r$ to fid_r
15:     **if** $epoch \mod p = 0$ **then**
16:         e1 = polyfit_slope(recon_losses[-$w$:], deg=1)
17:         e2 = polyfit_slope(s_scores[-$w$:], deg=1)
18:         e3 = polyfit_slope(fid_r[-$w$:], deg=1)
19:         e4 = polyfit_slope(fid_g[-$w$:], deg=1)
20:         **if** e1 > 0 and e2 > 0 and e3 > 0 and e4 > 0 **then**
21:             // stop with optimal latent dimensionality n_z
22:             $compressing \leftarrow 0$
23:         **else if** e2 >0 and $latent\_decrease$ ¿ 1 **then**
24:             $latent\_decrease \leftarrow 1$
25:         **end if**
26:     **end if**
27:     **if** $compressing = 1$ **then**
28:         $n_z = n_z$ - $latent\_decrease$
29:         model.update_latent_dim(new_dim=$n_z$)
30:     **end if**
31: **end for**

---

As stated in the previous subsection, we apply the decision mechanism for possible neuron pruning after each $p = 5$ training epochs. At each evaluation epoch $e$, we calculate the slope of the Silhouette score and reconstruction losses using a sliding window of 20 epochs. For each window, we calculate the first-degree polynomial using the least squares polynomial fit to obtain the slope of the curve. We use the slopes of the Silhouette score, reconstruction loss, FID for reconstructed images and FID for generated images to determine the stopping epoch $e$ after which we no longer reduce the latent dimensionality. The stopping epoch $e_s$ is thus defined as follows:

$$e_s = \min\{e|(S'(e) > 0) \wedge (R'(e) > 0) \wedge (FID'_r(e) > 0) \wedge (FID'_g(e) > 0)\} \quad (4)$$

where $(S', R', FID'_r, FID'_g)$ are the first-degree polynomial scores $p'$ (for the given metric curve) obtained through minimization of the squared error $E$:

$$E = \sum_{j=0}^{k} |p(x_j) - y_j|^2$$
$$p' = \arg\min_p \sum_{j=0}^{k} |p(x_j) - y_j|^2 \tag{5}$$

where $k$ is the size of the sliding window (we choose $k$=20), $x$ is the $x$-coordinate of each sample point, $y$ is the y-value of the given sample point for the given metric (Silhouette score, Reconstruction loss, and FID for reconstructed or generated images). In other words, we seek the first epoch for which the slopes of the observed metrics calculated over the last k=20 epochs are all positive. This means that FID and Reconstruction loss have already reached the optimal peak value and are starting to increase (i.e., deteriorate), while the Silhouette score is improving. For a different overview of our method, see Algorithm 1.

In the following section, we demonstrate the effectiveness of our proposed ALD-VAE on four standard vision datasets. We show that our adaptive algorithm can save a significant amount of computational time as it achieves similar results as if we would use extensive grid search to find the optimal latent space dimensionality.

## 4 EXPERIMENTS

To show the robustness and efficiency of our ALD-VAE approach, we compare its performance to grid search for 4 image datasets of different complexity (SPRITES, MNIST, FashionMNIST, EuroSAT, see Section 4.1) with known labels that enable evaluation of the clustering capability. For each dataset, we evaluate the model's performance on generative and reconstructive FID, Reconstruction loss and Silhouette score. We train the following models:

1. A new VAE model for different (fixed) latent dimensionalities (grid search).

2. The proposed ALD-VAE model with gradually compressing latent space.

We train all experiments on 5 different seeds with the same VAE architecture. The number of epochs as well as the used encoder and decoder networks are mentioned for each dataset. All encoder networks also include the two additional fully-connected layers to produce the $\boldsymbol{\mu}$ and $\boldsymbol{\sigma}$ vectors.

### 4.1 DATASETS DESCRIPTION AND VAE MODELS SPECIFICATION

The **SPRITES dataset** (Li & Mandt, 2018) contains colour images of animated characters (sprites). Each input is a sequence of 8 images showing random game characters performing one of 9 actions. For the encoder and decoder networks, we used a 3D convolutional layer adapted from VideoGPT (Yan et al., 2021). We train all models for 300 epochs, the fixed models with latent sizes 12, 16, 24 and 32 and ALD-VAE with initial latent sizes $n_z = 100, 80, 64$.

**MNIST** (Deng, 2012) is a widely used dataset comprising grayscale images with written digits. For the encoder and decoder networks, we use 2 fully-connected layers with 400 hidden dimensions. We train for 200 epochs with latent sizes 4, 8, 12, 16, 24 and 32 for the fixed dimensionality baseline scenarios and initial latent sizes $n_z = 100, 80, 64$ for our adaptive ALD-VAE scenario.

**FashionMNIST** (Xiao et al., 2017) contains 28x28 grayscale images of fashion articles from 10 different categories. For this dataset, we use the same encoder and decoder networks as for the MNIST dataset, i.e. 2 fully-connected layers with 400 hidden dimensions. We trained the models for 200 epochs and we compared the latent sizes 8, 16, 24, 32 and 64 in the fixed dimensionality experiments and the initial latent sizes $n_z = 100, 80, 64$ for ALD-VAE.

**EuroSAT** (Helber et al., 2019) contains 64x64 RGB satellite images showing pieces of land from 10 categories (e.g. *forest*, *river*, *industrial* etc.). For the encoder and decoder networks, we use 4x 2D convolutional layers with 256 hidden dimensions and SiLU activation functions. Compared to the other datasets, EuroSAT has a higher level of detail and variability in the images and is thus more difficult to reconstruct and even cluster. We trained the models again for 200 epochs and compared the latent sizes 16, 24, 32 and 64 in the fixed dimensionality experiments. The initial latent sizes for ALD-VAE were $n_z = 100, 80, 64$.

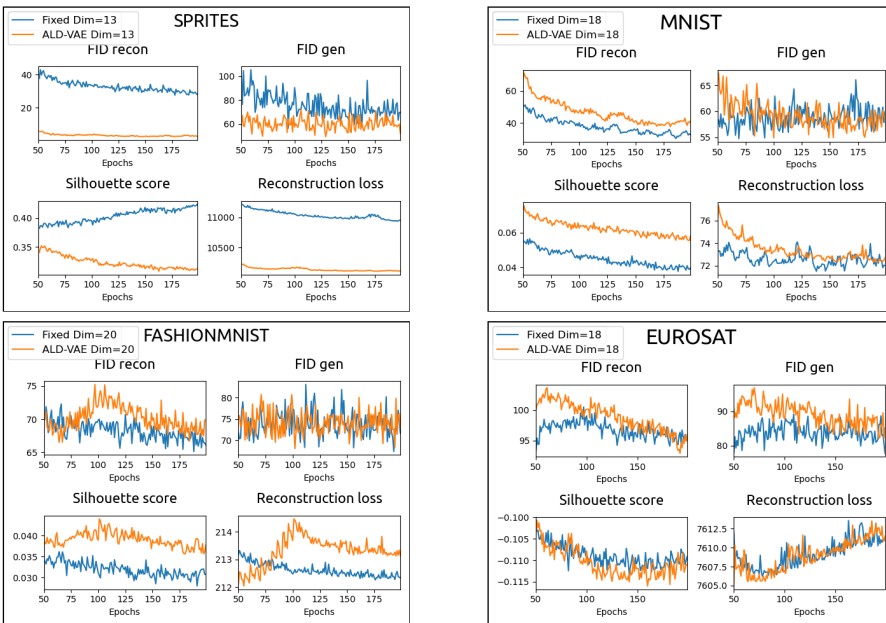

Figure 3: Comparison of the four metrics when using a fixed latent dimensionality for the whole training (blue curves) or our ALD-VAE latent space compression that converged to the same final dimensionality (orange curves). We show FID for reconstructed and generated images (lower is better), Silhouette score (higher is better) and Reconstruction loss (lower is better). Shown on four different datasets.

# 5 RESULTS

In this section, we show a comparison of the proposed ALD-VAE adaptive approach to the standard grid search approach on four standard vision datasets. We also show the effect of various initialization conditions and seeds.

## 5.1 COMPARISON WITH RANDOM SEARCH

First, we show in Fig. 2 the quality of the latent space dimensionality $n_z$ as detected online by the proposed ALD-VAE algorithm. To show the quality of the online detection, we calculated the corresponding scores (Silhouette score, FID, reconstruction losses) for the model as if it would decrease $n_z$ until $n_z = 2$, ignoring the stopping criterion, and visualise the point, where the ALD-VAE actually decided to stop the reduction of the latent space dimensionality. As can be seen in Fig.2, the ALD-VAE finds a balance between the four observed metrics. As expected, the algorithm stops for all the datasets at the minimum (or directly after) of FID and Reconstruction loss, right after Silhouette score started increasing.

Next, we compared the ALD-VAE algorithm with another model trained with a fixed dimensionality that is the same as the final optimal $n_z$ found by ALD-VAE. The results for the four datasets are shown in Fig. 3. On average, it took 50 epochs until the model converged to the optimal dimensionality with the exception of FashionMNIST, which took 100 epochs to converge. As seen in the plots, the training curves either converge to very similar values (e.g. for the EuroSAT dataset), or there is a trade-off between some of the metrics - e.g., better Silhouette score for ALD-VAE in FashionMNIST, but worse final reconstruction loss. However, the differences are in all cases marginal and the overall outcome is almost identical when training with ALD-VAE compared to the optimal dimensionality fixed from the beginning. There is thus no drawback to using the significantly faster ALD-VAE compared to hyperparameter grid search over multiple dimensionalities.

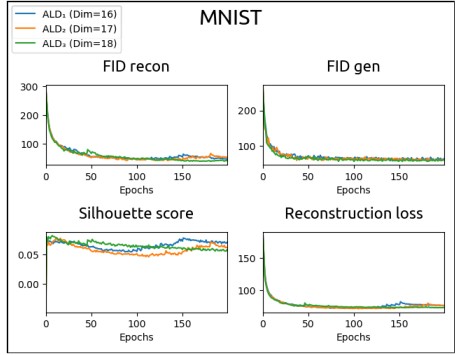 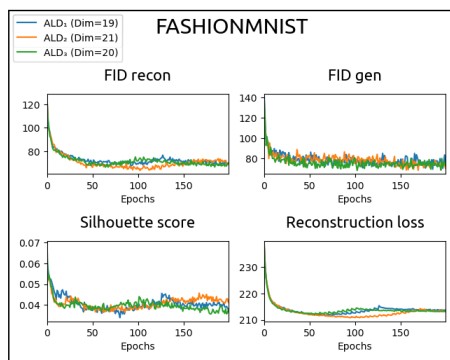

Figure 4: Training our ALD-VAE with different initial conditions. $ALD_1$ was trained with initial latent dimensionality $n_z = 100$, $ALD_2$ was trained with initial $n_z = 80$ and $ALD_3$ was trained with initial $n_z$=64. We show the results for the MNIST (left) and FashionMNIST (right) datasets. The $Dim$ values in brackets for each ALD-VAE show the final converged $n_z$.

## 5.2 VARIOUS INITIALIZATION CONDITIONS AND SEEDS

To see how robust our ALD-VAE is, we trained several models with various initialization conditions and various seeds. First, to see how the initial $n_z$ influences the final optimal dimensionality, we trained three models for each dataset. The first model had the initial $n_z = 100$, the second model had the initial $n_z = 80$ and the third model had the initial $n_z$=64. In Fig. 4, you can see results for MNIST and FashionMNIST. Both figures show the final optimal $n_z$ for each model. As you can see, the optimal dimensionalities only differ by 1-2 dimensions and the four observed metrics have very similar values. The results were similar also for the EuroSAT and SPRITES datasets. Note that we did not have to increase the number of training epochs for larger $n_z$ as the adaptive neuron pruning can speed up the initial latent compression based on the decreasing Silhouette score (see Section 3.3.1 for the mechanism description).

We also trained ALD-VAE on 5 different seeds for each dataset. While the resulting $n_z$ was relatively consistent for MNIST and FashionMNIST (standard deviation sd=3 dimensions for MNIST and sd=2 dimensions for FashionMNIST), there were more significant differences for the two remaining datasets (sd=6 for EuroSAT and sd=7 dimensions for SPRITES). However, we also found that the diverse final dimensionality $n_z$ was caused by a different course of the observed metrics in the particular seeds. It might be thus possible that, provided we are seeking a balance between the clustering and reconstructing capabilities, there are different optima in different seeds due to the noisy nature of VAEs.

## 6 DISCUSSION

In Section 5, we have shown that the latent space compression with ALD-VAE reaches the same performance as when training a model with fixed dimensionality from scratch while being significantly faster. We have also shown that our model is robust towards various initial $n_z$. Here we discuss further findings and possible extensions.

Firstly, we found that the optimal latent dimensionality obtained with ALD-VAE sometimes differs across different seeds. However, we observe that the evaluation metrics (Reconstruction loss, FID, Silhouette score) have also variable shapes during latent space compression. Considering that the optimum dimensionality was compared with the hyperparameter grid search in each seed, we attribute the differences across seeds to the stochastic and noisy nature of VAEs.

Next, the stopping mechanism of ALD-VAE was designed to find a balance between reconstruction and clustering quality. However, users have the flexibility to adjust the decision thresholds to favour one quality over the other, providing a customizable approach.

Finally, the experimental evaluation was performed on image datasets, allowing for the straightforward calculation of the Fréchet Inception Distance (FID). However, it is important to note that for datasets of other types, using FID might not be appropriate or may need to be replaced with

an alternative metric more suitable for the specific data. We leave it for future research to explore ALD-VAE, e.g., for text data or robotic actions.

# 7    CONCLUSION

We have introduced a new method for automatically determining the optimal size of the latent space in Variational Autoencoders during a single training. Compared to parameter grid search, our approach significantly speeds up the process of finding the optimal latent space dimensionality.

Our model, called ALD-VAE, gradually reduces the size of the latent space during the training process and evaluates the quality of the reconstruction and clustering using the FID score, Reconstruction loss and Silhouette score. We compare our method to the results obtained using a random hyperparameter search on four different image datasets and show that our proposed method can approximate the optimal size of the latent space and eliminate the need for manual tuning. Furthermore, ALD-VAE reaches comparable accuracies as the baselines trained with the same latent size from scratch. As ALD-VAE requires only simple adjustments to the original architecture and is easy to implement, it might be a convincing alternative to the classical models that require manual tuning of the latent space dimensionality.

ACKNOWLEDGMENTS

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
