# OpenReview forum: "Adaptive Compression of the Latent Space in Variational Autoencoders"
_ICLR.cc/2024/Conference — Submitted to ICLR 2024_

### Official Review · Reviewer_g7w5 · 2023-10-30

**Soundness:** 2 fair
**Presentation:** 2 fair
**Contribution:** 2 fair
**Rating:** 5
**Confidence:** 2

**Summary:**

The paper proposes a method that adaptively decreases the latent space size during the VAE training procedure. The stopping mechanism is based on Silhouette score, reconstruction loss, and FID.

**Strengths:**

- The writing is easy to follow and understand.

**Weaknesses:**

- Lack of comparison with other SOTA methods. e.g. MaskAAE(Mondal, Arnab Kumar, et al.), GECO(De Boom, et al.),
- Lack of ablation study. It might be worth ablating the contribution of e.g. Silhouette score for the decision rule. Are all of them really useful?
- There are no clear quantitative results. All comparisons of the four metrics are shown in Figure 3, and it’s not always easy to compare the two curves. (e.g. FID gen in MNIST, FashionMNIST, reconstruction loss in EUROSAT, etc.) A table might be more clear.

**Questions:**

The paper claims that the proposed method is significantly faster, but there is no analysis for time. Why is the proposed method faster? Can the proposed method converge with less epochs? The grid search might need to be run multiple times but it can be run in parallel, while the proposed method has to run clustering and needs to run in sequential order.

---

### Official Review · Reviewer_fzax · 2023-10-31

**Soundness:** 1 poor
**Presentation:** 1 poor
**Contribution:** 1 poor
**Rating:** 1
**Confidence:** 4

**Summary:**

The author proposes a method that automatically determines the optimal latent space dimensionality in Variational Autoencoders

**Strengths:**

The author proposes a method that automatically determines the optimal latent space dimensionality in Variational Autoencoders

**Weaknesses:**

1. This paper falls short of the standards required for publication. The novelty presented in the work is notably limited, with the proposed algorithm that render the research somewhat trivial. The method is basically about:
- train a model
- evaluate the model (use eq.4)
- reduce the dimension when the evaluation result does not converge.
- do everything again if the dimension is reduced.

  This dimension-reduction pipeline looks like some simple hyper-parameter(the dimension of the latent) search loop. How about just launching a dozen of training processes with different dimensionalities? Also, the evaluation methods seem trivial as well: FID, reconstruction loss, K-means. All of them were sufficiently discussed in Machine Learning communities, simply aggregating them yields limited novelty.

2. Furthermore, the absence of a baseline comparison is a big problem. Even some very basic method such as PCA is not included.

3. The paper's focus on dimension reduction **alone**, as a research topic in representation learning, might not be impactful. A more relevant and insightful approach would be to evaluate representations in terms of "bits per dimension", "ELBO", or "rate-distortion," rather than purely dimensionality. For example, [Alemi et al.](https://proceedings.mlr.press/v80/alemi18a/alemi18a.pdf) had some theoretical analysis of VAE's rate-distortion curve. [Higgins et al.](https://openreview.net/pdf?id=Sy2fzU9gl) discussed the disentanglement of the representation. [Balle et al.](https://arxiv.org/abs/1611.01704) proposed a method can use VAE for image compression, whose representation can be quantized and entropy-coded as bitstream. I also recommend referring to the paper at [Yang et al.](https://www.nowpublishers.com/article/Details/CGV-107) for a comprehensive introduction to data compression, which could provide valuable insights and context. Actually none of these work cares about the dimensionality of the representation, what people really want to know is "how much or what kind of information the representation can carry".

In conclusion, this paper falls short of the quality standards expected for a publication in ICLR. At least, more analysis and more baselines are needed.

**Questions:**

see above.

---

### Official Review · Reviewer_ksU9 · 2023-10-31

**Soundness:** 2 fair
**Presentation:** 3 good
**Contribution:** 2 fair
**Rating:** 5
**Confidence:** 4

**Summary:**

In the paper, the authors propose a method to automatically find the optimal latent dimension during the training of VAEs. This technique is based on the continuous analysis of the slopes of some metrics: the Silhouette score, the reconstruction loss, and the FID (both for reconstructed and generated data). The experiments performed on 4 real-world datasets (SPRITES, MNIST, FMNIST, and EuroSAT) show that the resulting ALD-VAE model (with latent dimension adjustment), compared to the classical VAE (with a fixed optimal size of the latent), provides (at convergence) comparable results for the mentioned metrics. In addition, training ALD-VAE on MNIST and FMNIST to find the optimal latent dimension is robust to different initial latent sizes and data seeds.

**Strengths:**

(1) The paper is clearly written and easy to understand.

(2) The proposed ALD-VAE model seems to be faster compared to VAE with grid search for optimal latent dimension.

**Weaknesses:**

(1) The significance of the proposed solution (although somewhat novel) was not sufficiently justified.

(2) Limited experimental setup (lack of experiments on more complicated large-scale datasets, e.g., CelebA).

(3) The authors emphasize that their solution is significantly faster than those using a grid search procedure (which is reliable), but they do not provide any experimental evidence for this claim.

(4) As the authors claim, the optimal latent dimensions obtained by ALD-VAE trained on the SPRITES and EuroSAT datasets are not robust to different data seeds.

**Questions:**

Minor comment: wrong sign '¿' in line 23 of Alg. 1.

---

### Official Review · Reviewer_iAX2 · 2023-11-01

**Soundness:** 2 fair
**Presentation:** 2 fair
**Contribution:** 2 fair
**Rating:** 5
**Confidence:** 4

**Summary:**

Variational AutoEncoders (VAEs) are powerful generative models, but the optimal latent space size is difficult to set. This paper proposes a heuristic method called ALD-VAE capable of finding the optimal latent dimension by gradually decreasing the latent size and evaluating the model performance during the training process. The method is compared to grid search and is shown to be faster and achieve the better latent space sizes on four image datasets: MNIST, FashionMNIST, SPRITES and EuroSAT.

**Strengths:**

- This paper proposes an heuristic approach to find the optimal latent dimension for VAEs. During VAE training, the method gradually decrease the latent space size by evaluating the model performances in FID score, Silhouette score, and reconstruction error. When the slopes of all these scores are positive, then the pruning of the latent space size stops and the method returns the optimal latent space size.

- The empirical results shown in Fig. 2 show that the method is able to find the latent space sizes that produce almost the lowest FID scores, the highest Silhouette scores, and the lowest reconstruction errors on four datasets.

**Weaknesses:**

- The algorithm is quite heuristic and straightforward. There is no theoretical justification of the method.

- There are too many hyper-parameters in this method, such as p = 5, n = 5, window w, number of clusters, latent_decrease and so on. The authors need to investigate how sensitive are those parameters and how different values of the hyper-parameters affect the final performance.

- The datasets used in this paper are small datasets, in terms of both number of images and image resolution. The authors need to do experiments on other large-scale high resolution image datasets such as CelebA-HQ, and ImageNet to evaluate the proposed method. The large-scale high resolution image datasets can also be used to evaluate how sensitive are those hyper-parameters.

**Questions:**

- The number of clusters in Silhouette score should also be chosen or optimized. How does the number of clusters affect the Silhouette score?

- During the pruning process, how do you decide which $n$ neurons are pruned? If you randomly pick $n$ neurons and remove them, will you remove the neurons that are more important than the remaining neurons?

- The stopping criteria is that "the slopes of the observed metrics calculated over the last k=20 epochs are all positive". I think this may not be robust. Could it be possible that in future epochs, the FIDs and reconstruction loss be decreasing again and the the Silhouette score be increasing? A related question is that the optimality in Eq. 2 is a multi-objective optimization problem. How to balance the three different objectives?

- There are several important parts not clear. FID'_r and FID'_g are not defined. What are polynomial scores? $p$ in page 5 two lines below Alg. 1 is a scalar 5, but in Eq. 5 is a function.

---

### Meta-Review · Area_Chair_6i7D · 2023-12-05

**Metareview:**

This paper proposes a heuristic method called ALD-VAE to find the optimal latent dimension. The idea is to gradually decrease the latent dimension by evaluating the model performance during the training process. Experiments are provided to compared the proposed approach with a number of baselines.

Reviewers are not convinced by the proposed approach, mainly due to:
1. Limited novelty as they view the method as the usual model selection process with a somewhat slightly novel selection criteria;
2. The datasets used in the experiments are not large-scale.

**Justification For Why Not Higher Score:**

No clear reason for acceptance, also no author feedback/revision available.

**Justification For Why Not Lower Score:**

N/A

---

### Decision · Program_Chairs · 2024-01-16

Reject